Power analyses to inform clutch sampling design to determine the breeding sex ratio in populations with multiple paternity

Quennessen Vic viquennessen@gmail.com 1 2
Fuentes Mariana M.B.P. 3
Komoroske Lisa 4
White J. Wilson 1 2
1 Coastal Oregon Marine Experiment Station , Newport , OR , United States of America
2 Department of Fisheries, Wildlife, and Conservation Sciences, Oregon State University , Corvallis , OR , United States of America
3 Department of Earth, Ocean and Atmospheric Science, Florida State University , Tallahassee , FL , United States of America
4 Department of Environmental Conservation, University of Massachusetts at Amherst , Amherst , MA , United States of America
Mata Fernando
Electronic publication date: 2025 Oct 28
Publication date: 2025
Volume: 13
Electronic Location ID: e20165
Received 2025 Feb 13; Accepted 2025 Sep 11
Copyright: ©2025 Quennessen et al.
Copyright year: 2025
Copyright holder: Quennessen et al.
License: This is an open access article distributed under the terms of the Creative Commons Attribution License, which permits unrestricted use, distribution, reproduction and adaptation in any medium and for any purpose provided that it is properly attributed. For attribution, the original author(s), title, publication source (PeerJ) and either DOI or URL of the article must be cited.
License URL: https://creativecommons.org/licenses/by/4.0/

Keywords: Multiple paternity, Power analyses, Chelonia mydas, Sampling design, Breeding sex ratio, Sea turtle, Paternal contribution mode, Operational sex ratio, Experimental design, Paternal genotype reconstruction

Funding: National Science Foundation OCE-1904818 1904439 1904615 Sea Grant/National Marine Fisheries Ecosystem and Population Dynamics Ted Thorgaard Student Research Fund This work was supported by the National Science Foundation (OCE-1904818, 1904439, 1904615). VQ was funded by the Sea Grant/National Marine Fisheries Ecosystem and Population Dynamics graduate fellowship and the Ted Thorgaard Student Research Fund. The funders had no role in study design, data collection and analysis, decision to publish, or preparation of the manuscript.

==============================
In many species, demographic assessments of population viability require an estimate of the number or proportion of breeding adults in a population that are male (the breeding sex ratio). However, this estimate is often difficult to obtain directly in species with multiple paternity when males are difficult to sample. Parentage analysis of breeding females and offspring can produce this estimate by identifying the number of unique males that contribute genetic information to (i.e., sired) a given cohort. There is an added challenge of choosing a sample design with the desired level of confidence to identify all the fathers contributing to a cohort, either at the scale of individual clutches or an entire nesting season, given limited resources. Sampling effort can be defined as the number of offspring sampled per clutch, or the number of clutches sampled per breeding season, depending on the analysis. The minimum number of samples required may depend on the proportions of eggs that different fathers fertilize in a clutch (the paternal contribution mode), the total number of fathers fertilizing a clutch, the proportion of adults available for breeding that are male (the operational sex ratio), and population size. We conducted power analyses to quantify the confidence in identifying all fathers in animal populations with multiple paternity. We simulated sampling a theoretical sea turtle population with a range of population demographics, mating systems, and sampling effort, and used the proportion of 10,000 simulations in which all fathers were identified as a proxy for confidence. At the clutch level, confidence was strongly dependent on the paternal contribution mode, and when it was skewed, it also depended on the total number of fathers contributing and the number of offspring sampled. However, sampling about one third of a clutch was sufficient to identify all fathers with high confidence for most scenarios, unless the paternal contribution mode was extremely skewed and there were many contributing fathers, such that some fathers fertilized very few eggs and were difficult to detect. At the scale of an entire nesting season, confidence was more strongly affected by the operational sex ratio, the proportion of clutches sampled, and the presence or absence of polygyny than by the lesser effects of paternal contribution mode and within-clutch sample size. Sampling fewer offspring from more clutches increased confidence compared to sampling more offspring from fewer clutches. Relaxing the minimum required proportion of fathers identified from 100% to 90% led to high confidence while sampling 50% to a maximum of 75% of clutches, depending on the mating system, even as the population size increased by an order of magnitude. Our approach and results can be widely informative for sample design as well as quantifying uncertainty in existing and future estimates of the number of breeding males in populations with multiple paternity.

Introduction

Efficient field sampling requires balancing limited resources with the collection of sample sizes large enough to accurately estimate demographic rates and values. Two such values are the operational sex ratio (the proportion of adults that are available to breed that are males) and the breeding sex ratio (the proportion of adults that successfully breed that are males) (Emlen, 1976; Bensch et al., 1999) which ultimately contribute to the effective population size, extinction risk, and overall population viability (Sugg & Chesser, 1994; Holman & Kokko, 2013; Veliz et al., 2016). These quantities can be vital for accurately modeling and predicting population dynamics to inform species management and conservation. Unfortunately, sampling males is often overlooked in favor of focusing on females or offspring, which may be easier to sample, or which are thought to be the more important state variable for population dynamics (Carr, 1980; Whiting et al., 2021; Wallace et al., 2025), which reduces accuracy in modeling population dynamics and may impede conservation goals (Engen, Lande & SÆther, 2003; Alonzo et al., 2008; National Research Council US, 2010; Gerber & White, 2014; Piacenza, Richards & Heppell, 2019). This is especially true for populations where males have nonlinear effects on population dynamics, are not able to successfully mate due to endocrine disrupting compounds, or have been reduced to low levels in the population (Rankin & Kokko, 2007; Miller & Inouye, 2011; Refsnider & Janzen, 2016; White et al., 2017). Unfortunately, for species with long lifespans like sea turtles, waiting until skewed offspring sex ratios result in decreases in reproductive success will result in much longer recovery times (Komoroske et al., 2017; Hays et al., 2023). An added complication is that certain species’ mating systems make it difficult to determine how those males contribute to reproduction and population dynamics. For example, many species exhibit polygamy—a mating system with polygyny (males mate with multiple females), polyandry (females mate with multiple males), or both (Emlen & Oring, 1977; Taylor, Price & Wedell, 2014; Tarka et al., 2018). In those species, breeding success may vary greatly among males, so the breeding sex ratio would be very different from the operational sex ratio and difficult to determine by simply censusing the number of males in the population (Lindström & Kokko, 1998; Veliz et al., 2016).

Many populations of sea turtles (superfamily Chelonioidea) exhibit polygamy, especially multiple paternity (Lee et al., 2018; Hays, Shimada & Schofield, 2022; Lee & Hays, 2024), making them an excellent case study to quantify the challenge of estimating the breeding sex ratios given limited sampling resources. This is especially important given that sea turtles are thought to be particularly vulnerable to climate change, with increasingly female-biased sea turtle populations as climate change progresses that may lead to reduced reproductive success, reduced effective population size, and genetic drift (Girondot et al., 2004; Jensen et al., 2018; Patrício et al., 2021; Hays et al., 2023; Lee & Hays, 2024; Wallace et al., 2025; Hays, Laloë & Seminoff, 2025). Unfortunately, observing every copulation attempt in a sea turtle population is difficult and resource-intensive, copulation does not always lead to fertilization, and males do not come ashore or provide parental care; altogether, these make it harder to determine the number of males contributing to reproduction in a sea turtle population (Limpus, 1993; Komoroske et al., 2017; Hays, Shimada & Schofield, 2022). Consequently, many studies estimating and predicting sea turtle population dynamics focus on nesting females and offspring (either eggs or hatchlings, alive or dead), without including males, immature females, or both, which impairs the accuracy of these estimations (Butler, 1987; Chaloupka & Limpus, 2001; Chaloupka et al., 2008; National Research Council US, 2010; Piacenza, Richards & Heppell, 2017). Genetic analyses using tissue samples from field sampling of nesting females and offspring can help to reconstruct otherwise unavailable paternal genotypes through kinship analyses (Dewoody et al., 2000; Neff, Repka & Gross, 2000). However, for sea turtles as well as many fish, invertebrate, and amphibian species with many large clutches, it is prohibitively expensive to sample all offspring, even for small populations (Myers & Zamudio, 2004; Veliz et al., 2016; Scheelings, 2023). Ergo, representative individuals must be sampled strategically in order to maximize the number of fathers that can be confidently identified without using more effort or resources than needed.

Here, we address this issue of sample design to estimate the number of breeding males (fathers, from now on) using simulations for power analysis. Our key questions were, first, how large of an offspring sample size is needed to identify all fathers that fertilize eggs in a single clutch with high confidence? Second, for an entire sea turtle breeding season, does sampling more clutches with a smaller offspring sample size or sampling fewer clutches with a larger offspring sample size result in a higher confidence of identifying all of the fathers contributing to that season? We answered these questions by simulating alternative sampling designs for hypothetical sea turtle populations that were based on the observed demographics of a Brazilian green sea turtle population. Using these results, we present field sampling recommendations to more accurately estimate the total number of reproductive males, improving projections of population dynamics to most efficiently use available effort and resources. We also present analyses to quantify uncertainty in existing and future estimates of fathers contributing to a clutch. This will in turn inform management and conservation efforts for populations with multiple paternity and males that are hard to sample.

Materials & Methods

We simulated scenarios in which we sampled a population of nesting females where we assumed a static pool of adults available to breed (operational population size) and a specified operational sex ratio, expressed as the ratio of available males to all available adults, i.e., the proportion male. Females were able to mate with multiple males and nest multiple times in a single season, such that paternity could be shared between up to five males for clutches laid by a single female. For simulations with polygyny, males were also allowed to mate with up to five females, such that they could be represented in the clutches of up to five females. Data to parameterize the model, including distributions of clutches per female and offspring per clutch, were based on a green sea turtle subpopulation nesting on Praia do Leão on Fernando de Noronha (FdN), Brazil (Bellini, 1996; parameters in Table 1; field data in Table S1). All field work and data collection, including animal capture and tagging, was performed following protocols approved by the Institutional Animal Care and Use Committee at Florida State University (permits 1803, and PROTO202000076) and the Brazilian Ministry of Environment (MMA), Chico Mendes Biodiversity Conservation Institute (ICMBio), and Biodiversity Authorization and Information System (SISBIO, approval 69389-12).

Table 1 Demographic and model parameters.

	Parameter	Value	
Demographic parameters	Clutches per mother	∼Nμclutch=4.95,σclutch=2.09
source: field data (Table S1)	
Eggs per clutch	∼Nμeggs=100.58,σeggs=22.61
source: field data (Table S1)	
Decreasing probabilities of mating with 1–5 mates	ℙ = (0.463, 0.318, 0.157, 0.034, 0.028)
source: based on multiple paternity
rates for female green sea turtles (Table S2)	
Uniform probabilities of mating with 1–5 mates	ℙ = (0.2, 0.2, 0.2, 0.2, 0.2)	
No probability of mating with more than 1 mate	ℙ = (1, 0, 0, 0, 0)	
Model parameters	Operational population size	100, 200, 500, 1,000	
Number of offspring to sample within clutches	1–96	
Number of offspring to sample within clutches across a season	32, 96	
Number of simulations	10,000	

We assumed that sampling did not cause adult mortality or affect reproductive parameters, that there was no sampling error that would negatively affect the success of any subsequent genetic analyses, and that post-hoc genetic analyses found all contributing males (fathers) for sampled offspring, such that our analysis was only concerned with sampling effort. We performed computational simulations to model sampling single clutches (within-clutch analysis) or across clutches for an entire nesting season (within-season analysis, Fig. 1) to quantify sampling efforts that identified all fathers with a high confidence of 80% or higher. For both within-clutch and within-season analyses, we used the proportion of 10,000 simulations where we identified all contributing fathers to represent our confidence.

Figure 1 Schematic of the within-season model framework.

From left to right, the figure simulates (A) building a male breeding pool vector, (B) breeding, (C) nesting, and (D) sampling of the clutches and the males that fertilized them. Red circles represent males, blue triangles represent females, and purple squares represent clutches, with numbers representing the fathers that fertilized that clutch. This example assumes an operational population size of 100, an operational sex ratio of 0.05, 5 available males, 95 available females, declining probabilities of breeding with one to five mates, and 20% of clutches sampled (sampled clutches are outlined in black). Females 7 through 95 did not mate in this scenario as there were no males left in the male breeding pool vector after female 6 mated with male 5. Because 6 females successfully mated, the sample size of 20%, or 5 clutches out of 25, was not enough to identify all males, because the clutch from female 6 was not sampled and was the only clutch sired by male 5.

We capped the number of potential fathers for a clutch and for breeding with a single female at five (Table 1) because of a lack of evidence in the primary literature for multiple paternity in wild green sea turtles with more than five fathers (Fitzsimmons, 1998; Lee & Hays, 2004; Ekanayake et al., 2013; Purnama, Zamani & Farajallah, 2013; Alfaro-Núñez, Jensen & Abreu-Grobois, 2015; Chassin-Noria et al., 2017; Türkozan et al., 2019, Table S2). The probabilities of adults mating with one to five mates either decreased (‘decreasing polyandry’ and ‘decreasing polygyny’) or were uniform (‘uniform polyandry’ and ‘uniform polygyny’), with specific probability distributions outlined in Table 1. We also included simulations that did not include polygyny, due to a lack of data robustly quantifying polygyny in sea turtles, but see (Crim et al., 2002; Stewart & Dutton, 2014; Natoli et al., 2017; Gaos et al., 2018; Türkozan et al., 2019; Horne et al., 2023; Vella & Vella, 2024; Amorim et al., 2025; Bispo et al., 2025). For the within-season analysis, sample sizes of offspring per clutch were based on the physical constraint of 96-well plates, which are often used for genetic sequencing, such that a single plate could hold one set of 96 samples or three sets of 32 samples. In relation to the average clutch size we modeled of (100.58), these translate to almost all and roughly one third of the offspring in a clutch.

One source of uncertainty about sea turtles’, and many other species’, reproductive biology is the paternal contribution mode–the proportions of eggs fertilized by the different fathers of a clutch. In our simulations, we therefore explored a range of different biologically plausible paternal contribution modes: Random, Exponential, Dominant 50, Dominant 70, Dominant 90, and Mixed Dominant. Under the Random mode we assumed that each father had the same probability of fertilizing eggs within the clutch. For the Exponential mode, we assumed a hierarchy of fathers, such that one fertilized half the eggs, the next fertilized one quarter of the eggs, the next fertilized one eighth of the eggs, and so on, with the last father fertilizing as many eggs as the father before. Under the Dominant modes, we assumed that there was one dominant father who fertilized 50% (Dominant 50), 70% (Dominant 70), 90% (Dominant 90), or any one of these values (Mixed Dominant) of the eggs in the clutch, with the other non-dominant fathers fertilizing the remaining 50%, 30%, or 10% of the eggs randomly. For sea turtles specifically, the Mixed Dominant mode is likely the most realistic paternal contribution mode tested here; genetic analyses typically find that a dominant male fertilizes between 50 and 90% of eggs in a given clutch (Fitzsimmons, 1998; Hoekert et al., 2002; Lee & Hays, 2004; Zbinden et al., 2007; Ekanayake et al., 2013; Alfaro-Núñez, Jensen & Abreu-Grobois, 2015; Chassin-Noria et al., 2017; Türkozan et al., 2019; Dolfo et al., 2025). However, all of the proposed paternal contribution modes have been recorded in studies of other animal species, such as the Random mode in dumpling squid, Euprymna tasmanica (Squires et al., 2014); the Dominant mode in red-sided garter snakes, Thamnophis sirtalis parietalis (Friesen et al., 2014), green swordtails, Xiphophorus helleri (Tatarenkov et al., 2008), and brachyuran crabs (Veliz et al., 2016); and a mix of Random, Exponential, and Dominant modes in veined Rapa whelk, Rapana venosa (Xue, Zhang & Liu, 2014).

Within-clutch analysis

To calculate the confidence of identifying all fathers within a single clutch, we modeled sea turtle reproduction with demographic stochasticity in the number of eggs per clutch and how the eggs were fertilized by multiple males. First, we simulated a clutch with a number of eggs drawn from a normal distribution with mean μeggs and standard deviation σeggs (Table 1), then rounded to the nearest integer. For each clutch, we assigned eggs individually to fathers based on draws from a distribution of potential fathers with weighted probabilities based on each paternal contribution mode (Table 2). For example, for the Dominant 50 mode and five fathers, the eggs were assigned to fathers one through five using weighted probabilities of 0.5, 0.125, 0.125, 0.125, 0.125 where the first (dominant) father fertilized about 50% of the eggs, and the remaining 50% were split randomly among the remaining four (non-dominant) fathers. Then, we sampled the clutch with a given sample size (or the whole clutch, if there were fewer total offspring than the specified sample size) to determine how many fathers were represented and could be identified. We performed 10,000 simulations for each combination of number of fathers and paternal contribution modes; exploratory analyses indicated that this was sufficient to obtain consistent results.

Table 2 Patterns of egg fertilization by paternal contribution mode.

Father contributions to fertilization show weighted probabilities by paternal contribution mode within a clutch as a function of how many total fathers there are. The marginal father contribution column shows the proportion of eggs fertilized by the least contributing, or least dominant, father, assuming that there are F = 5 total fathers contributing to the clutch.

Paternal contribution mode	Paternal contributions given F total fathers	Marginal paternal contribution forF = 5	
Random	1/F for each of F fathers	0.2	
Exponential	0.5from 1 to F−1 and 0.5F−1 for the last father	0.0625	
Dominant 50	0.5 for first father 0.5/(F-1) for each remaining father	0.125	
Dominant 70	0.7 for first father 0.3/(F-1) for each remaining father	0.075	
Dominant 90	0.9 for first father 0.1/(F-1) for each remaining father	0.025	
Mixed Dominant	Draw weighted contributions randomly from the Dominant 50, Dominant 70, and Dominant 90 paternal contribution modes	Either 0.125, 0.075, or 0.025	

Model parameters that we modified included the number of offspring that were sampled from each clutch (1–96; or the whole clutch, if there were fewer offspring than the sample size), the total number of fathers (one to five), and the paternal contribution mode (Random, Exponential, Dominant 50, Dominant 70, Dominant 90, or Mixed Dominant).

For each combination of paternal contribution mode, sample size, and actual number of fathers, we calculated the probability of identifying one to all of the contributing fathers within a clutch as the proportion of simulations where that many fathers were identified (reported in Table S3). We used those probabilities and Bayes’ Theorem (Eq. 1) to calculate the conditional probabilities of the actual number of contributing fathers given the number of fathers identified across paternal contribution modes (Table 3), where X represents the number of fathers that actually contributed and Y represents the number of fathers that were identified: (1) PXcontributed|Yidentified=PYidentified|Xcontributed∗PXcontributedPYidentified.

Table 3 Conditional probabilities of total number of fathers contributing to a clutch given the number of fathers identified.

We did not include situations with five identified fathers, as that automatically implies there were five total fathers, the maximum we allowed in these simulations. Some rows do not appear to add up to 1.000 due to rounding.

Paternal contribution mode	Probability of F fathers contributing given a sample size of 32	Fathers identified	Probability of F fathers contributing given a sample size of 96	
	F = 1	F = 2	F = 3	F = 4	F = 5		F = 1	F = 2	F = 3	F = 4	F = 5	
Random	1	0	0	0	0	1	1	0	0	0	0	
–	1	0	0	0	2	–	1	0	0	0	
–	–	1	0	0	3	–	–	1	0	0	
–	–	–	0.996	0.004	4	–	–	–	1	0	
Exponential	1	0	0	0	0	1	1	0	0	0	0	
–	1	0	0	0	2	–	1	0	0	0	
–	–	0.957	0.027	0.016	3	–	–	1	0	0	
–	–	–	0.805	0.195	4	–	–	–	0.990	0.010	
Dominant 50	1	0	0	0	0	1	1	0	0	0	0	
–	1	0	0	0	2	–	1	0	0	0	
–	–	0.991	0.008	0.000	3	–	–	1	0	0	
–	–	–	0.948	0.052	4	–	–	–	1	0	
Dominant 70	1	0	0	0	0	1	1	0	0	0	0	
–	0.986	0.010	0.002	0.001	2	–	1	0	0	0	
–	–	0.883	0.090	0.027	3	–	–	0.999	0.001	0	
–	–	–	0.772	0.228	4	–	–	–	0.992	0.008	
Dominant 90	0.879	0.030	0.030	0.031	0.030	1	0.998	0	0	0	0	
–	0.567	0.187	0.133	0.113	2	–	0.957	0.026	0.010	0.006	
–	–	0.437	0.308	0.255	3	–	–	0.827	0.122	0.051	
–	–	–	0.477	0.523	4	–	–	–	0.736	0.264	
Mixed Dominant	0.956	0.011	0.011	0.011	0.011	1	0.999	0	0	0	0	
–	0.799	0.088	0.062	0.051	2	–	0.986	0.008	0.003	0.002	
–	–	0.729	0.158	0.113	3	–	–	0.937	0.045	0.018	
–	–	–	0.775	0.225	4	–	–	–	0.900	0.100	

Within-season analysis

To calculate the confidence of identifying all fathers across multiple clutches in a nesting season, we modeled an entire breeding and nesting season with stochasticity in the assignment of fathers to offspring and sampling, and demographic stochasticity in the number of mates for either sex; the number of clutches laid; and the number of offspring per clutch.

First, we simulated an operational pool of potential parents of fixed size, with females (mothers) breeding with males (fathers) and nesting, and then the subsequent sampling (Fig. 1). We assumed an operational population size and operational sex ratio, which defined the total number of males and females available for breeding for the season (Table 1). We created a breeding pool of potential fathers, such that each male was able to breed with one to five females as randomly sampled with weighted probabilities from the appropriate multinomial probability distribution ℙ (Table 1); for simulations without polygyny, all the probability mass in ℙ was on 1 (i.e., each father only mated once). Then, we modeled each potential mother breeding with one to five unique potential fathers sampled from the breeding pool with weighted probabilities drawn from ℙ, until the pool ran out of potential fathers. If there were no more potential fathers available for breeding, the females remaining did not successfully mate or nest. For each mother that successfully mated, we simulated a number of clutches drawn from a normal distribution with mean μclutch and standard deviation σclutch, rounded to the nearest integer (Table 1). For each clutch, we simulated an integer number of eggs drawn from a normal distribution with mean μeggs and standard deviation σeggs, rounded to the nearest integer (Table 1). For each egg, we assigned a father by sampling the potential fathers with weighted probabilities based on the appropriate paternal contribution mode (Table 2). We tracked the clutches that were laid throughout the simulated season, and the fathers that fertilized them. Finally, we sampled a given sample size of offspring from a given proportion of clutches and, for each clutch, determined how many contributing fathers were identified based on the probabilities calculated from our within-clutch analysis (Table S3). Across the sampled clutches, we compared the cumulative total number of unique fathers identified to the total number of fathers that bred in that simulation to determine how many we identified.

The model parameters that we modified were the operational sex ratio (varied from 0.05–0.95), the proportion of clutches that were sampled (varied from 0.05 to 1.00, rounded up to the nearest integer clutch), the number of offspring that were sampled from each clutch (32; 96; or the whole clutch, if there were fewer offspring than the sample size), the paternal contribution mode (Random, Exponential, Dominant 50, Dominant 70, Dominant 90, or Mixed Dominant), and the mating system, with every combination of different levels of polyandry (decreasing polyandry and uniform polyandry) and polygyny (decreasing polygyny, uniform polygyny, and no polygyny, Table 1). We also modeled alternative simulations with different operational population sizes (200, 500, and 1,000), and a lower minimum required proportion of fathers to be identified to count as “successful” (90%).

We performed all analyses in R version 4.2.3, “Shortstop Beagle” (R Core Team, 2023), and all codes used to produce simulation outputs are publicly available as an R package on GitHub (vquennessen, 2025).

Results

Within-clutch analysis

When we simulated sampling offspring from individual clutches, our confidence in identifying all contributing males (fathers) increased with sample size and decreased with total number of fathers, regardless of paternal contribution mode (Fig. 2). With a sample size of 32 offspring per clutch, all fathers were confidently identified in most cases, except for very skewed mating systems with one dominant father and many total fathers contributing. The only case in which a sample size of 96 offspring per clutch failed to confidently identify all fathers was the Dominant 90 mode with five total fathers (Fig. 2E). The confidence also decreased as the marginal paternal contribution, or the smallest proportion of eggs fertilized by a single father, decreased (Fig. 3). The Random mode had the highest marginal contributions and the highest confidence, followed by the Dominant 50 mode, the Exponential mode, the Dominant 70 mode, and finally the Dominant 90 mode, which had the lowest marginal contributions and the lowest confidence (Figs. 2 and 3).

Figure 2 Proportion of simulations that identified all fathers for a clutch by paternal contribution mode and number of fathers.

Curves show the proportion of 10,000 simulations in which the total number of fathers was correctly identified, with the horizontal dotted line indicating a threshold of 0.80. Line color indicates the actual total number of fathers. Vertical dotted lines denote sample sizes of 32 and 96 offspring. Panels show results from different paternal contribution modes: (A) Random, (B) Exponential, (C) Dominant 50, (D) Dominant 70, (E) Dominant 90, (F) Mixed Dominant.

Figure 3 Marginal contributions of the least dominant male.

The height of the bars indicates the expected proportion of eggs that are fertilized in a clutch by the father that fertilizes the fewest eggs, i.e., the marginal contribution. Bars are clustered by paternal contribution mode, and the color of the bars indicates the number of total fathers, with increasing numbers of fathers left to right within each cluster. The Mixed Dominant paternal contribution mode is not included as it is a combination of the other dominant paternal contribution modes and averages out to the Dominant 70 mode results.

The conditional probabilities for how many fathers contributed to a clutch given how many were identified followed a similar trend. For the Random mode, the number of total fathers matched the number of fathers identified at least 99.6% of the 10,000 simulations with a sample size of 32, and for every simulation with a sample size of 96 (Table 3). The Dominant 50 mode was second most accurate, with at least 94.8% of simulations identifying all the fathers with a sample size of 32, and all simulations with a sample size of 96. These were followed by the Exponential mode (at least 80.5% and 99.0%), the Dominant 70 mode (at least 77.2% and 99.2%), and the Mixed Dominant mode (at least 72.9% and 90%), where a sample size of 96 always outperforming the sample size of 32 (Table 3). The Dominant 90 mode had the lowest accuracy, with as little as 43.7% of simulations identifying all fathers with a sample size of 32 and 73.6% with a sample size of 96 (Table 3).

Within-season analysis

When we simulated sampling clutches throughout an entire nesting season to determine the total number of contributing fathers, we found that the proportion of simulations where all fathers were identified increased with the proportion of clutches that were sampled and peaked at extremely low and extremely high operational sex ratios (Fig. 4). Differences in results between sampling 32 and 96 offspring per clutch were minimal, even for simulations with the most extreme paternal contribution mode, the Dominant 90 mode (Fig. 4, Fig. S1). For more moderate operational sex ratios, higher proportions of clutches needed to be sampled in order to maintain a confidence of at least 80% (Fig. 4). In the worst-case scenario—simulations with an operational sex ratio of 0.7, the Dominant 90 mode, uniform polyandry, and no polygyny—more than 90% of clutches had to be sampled in order to confidently identify all fathers (Fig. 4). A consistent difference in the proportion of clutches sampled of about 0.1 was required to maintain the same confidence between the Random and Dominant 90 modes, keeping all other parameters equal (Fig. 4). Simulations without polygyny required higher proportions of clutches to be sampled compared to simulations with decreasing polygyny, which required higher proportions of clutches to be sampled compared to simulations with uniform polygyny. Simulations with uniform polyandry required slightly more clutches to be sampled compared to simulations with decreasing polyandry (Fig. 4).

Figure 4 Confidence in identifying all fathers by mating system and paternal contribution mode when 32 offspring are sampled from each clutch.

Color indicates the upper edge of the confidence band, with solid fill representing simulations with the Random paternal contribution mode and contour curves representing simulations with the Dominant 90 paternal contribution mode. Within each panel, the x-axis represents the proportion of clutches across the entire nesting season that were sampled, and the y-axis represents the operational sex ratio. Columns of panels show different distributions of polyandry, with results from simulations with decreasing probabilities (“Decreasing polyandry” in (A), (B), and (C)) and uniform probabilities (“Uniform polyandry”, in (D), (E), (F)) of females breeding with one to five males. Rows of panels show different distributions of polygyny, with results from simulations with decreasing probabilities of mating with 1–5 females (“Decreasing polygyny”, (A), (D)), uniform probabilities of mating with 1–5 females (“Uniform polygyny”, (B), (E)), and zero probabilities of males breeding with more than one female (“No polygyny”, (C), (F)).

Figure 5 Proportion of simulations that successfully identify enough fathers by population size and threshold of fathers identified with uniform polyandry and uniform polygyny.

Color indicates the proportion of simulations with offspring sample sizes of 32 where the minimum required fathers were correctly identified, with solid fill representing simulations with the Random paternal contribution mode and contour curves representing simulations with the Dominant 90 paternal contribution mode. Within each panel, the x-axis represents the proportion of clutches across the entire nesting season that were sampled, and the y-axis represents the operational sex ratio. Columns of panels show different minimum proportions of fathers to ID to be counted as a “success”: 90% of all fathers (A, B, C, D) and 100% of fathers (E, F, G, H). Rows of panels show different starting population sizes of adults available for breeding: 100 (A, E), 200 (B, F), 500 (C, G), and 1,000 (D, H). The mating system modeled had uniform probabilities of mating with one to five mates (“uniform polyandry” and “uniform polygyny”).

These results were dependent on the size of the initial pool of adults available for breeding, or operational population size, as well as the proportion of fathers that needed to be identified for the simulation to be considered “successful”. As the operational population size grew, the higher the proportion of clutches that had to be sampled to confidently identify all fathers (Fig. 5). However, if identifying 90% of fathers or more was considered “successful”, then as few as 50% of clutches needed to be sampled for female dominated populations, and no more than 75% of clutches ever, even as the operational population size grew by an order of magnitude (Fig. 5). Even for mating systems without polygyny, no more than 75% of clutches need to be sampled to confidently identify 90% or more of fathers, even with the most skewed paternal contribution mode of Dominant 90 (Fig. S2).

Discussion

We simulated reproductive and sampling behavior to quantify the range of probabilities of successfully identifying all fathers contributing to a cohort of sea turtle offspring, both at the scale of individual clutches and an entire nesting season. At the individual clutch level, we found that a smaller sample size of about one third of offspring (compared to almost the whole clutch) was sufficient to confidently identify all fathers for most mating systems modeled; the only exception was for those scenarios with many fathers and a very skewed paternal contribution mode. The confidence in identifying all fathers increased with the number of offspring sampled in a clutch and marginal contributions, and therefore decreased with the total number of fathers and skew in the paternal contribution mode, which both reduce marginal contributions, in line with previous simulation analyses of salamander (Myers & Zamudio, 2004), fish (Dewoody et al., 2000), and sea turtle clutches (Chassin-Noria et al., 2017). Broadly speaking, sampling approximately one third of the offspring in a clutch should allow researchers to identify all fathers that contributed with high confidence. Previous analyses have found sampling twenty offspring in a clutch to be sufficient to identify up to five fathers, but based on only partially sampled clutches, and an average father fertilizing 34% of the eggs in a clutch, indicating a low level of skew in paternal contribution mode (Chassin-Noria et al., 2017). For those mating systems where marginal contributions are suspected to be small, estimates of confidence can be calculated as conditional probabilities based on the number of fathers identified and the potential paternal contribution modes considered (Eq. 1, Table 3).

At the nesting season level, our confidence in identifying all fathers increased with the proportion of clutches sampled, which had to increase in simulations with a moderate operational sex ratio to maintain the same level of confidence, peaking at operational sex ratios of 0.5 to 0.9. We found that increasing the sample size of offspring in a clutch minimally increased confidence. As paternal contribution mode skew increased, confidence decreased, and this effect of skew was stronger in simulations without polygyny. Increasing the sample size from one third to most of the offspring per clutch or having a paternal contribution mode with low skew did not affect confidence nearly as much as changes in the proportion of clutches sampled and the operational sex ratio. This is in line with a similar analysis based on detecting multiple paternity in brachyuran crabs, where sampling fewer offspring from more mothers led to higher statistical power in quantifying breeding males at the population level compared to sampling more offspring from fewer mothers (Veliz et al., 2016). For populations without polygyny, this is especially important, as the clutches sampled have to include at minimum one clutch from each nesting female to be able to successfully identify all the fathers, also necessitating smaller offspring sample sizes within clutches given limited resources. For simulations with larger operational population sizes, higher proportions of clutches had to be sampled to maintain high confidence. However, when the minimum required proportion of fathers identified was reduced from 100% to 90%, then it was possible to identify most fathers with high confidence by sampling as few as 50% of clutches for simulations with the best-case scenario—uniform polyandry, uniform polygyny, and the Random mode—and by sampling 75% of clutches across all scenarios. Notably, these results were consistent across mating systems, and are therefore applicable to a wide array of polygamous animal populations.

Previous studies estimating the number of breeding male sea turtles at the clutch level or for other species at the population level have often focused on simply detecting multiple paternity (Phillips et al., 2013; Stewart & Dutton, 2014; Veliz et al., 2016) as opposed to counting the number of contributing fathers, with many focused at the clutch level instead of extrapolating to the population level (but see Veliz et al., 2016). Many sea turtle analyses used very small sample sizes (as low as four hatchlings in a clutch or fewer than 1% of clutches), even while acknowledging that one dominant father often fertilized most of the eggs in a clutch (80% or higher) such that other fathers would fertilize very few eggs (Phillips et al., 2013; Tedeschi et al., 2014; Gaos et al., 2018; Türkozan et al., 2019). Studies that have conducted power analyses to estimate breeding sex ratios based on sample design have been in eusocial insects with haplo-diploid mating systems, like ants, bees, and wasps (Pedersen & Boomsma, 1999; Tarpy & Nielsen David, 2002), or plants that can self-pollinate (Christopher et al., 2019; Gibson et al., 2020) at the population level, and simulated populations of fish at the clutch level (Dewoody et al., 2000). This is therefore the first study to extend such power analyses to strategically calculate confidence in identifying all fathers at both the clutch and nesting season level for sexually reproducing animal populations with multiple paternity and demographic stochasticity, as well as the combined effects of different mating systems, sampling design, operational sex ratios, operational population sizes, and threshold proportions of fathers to identify to count as a “success”.

Given the scope of this project and the general availability of data, we were not able to account for all potential sources of variation. For example, operational and breeding sex ratios may vary within and between breeding seasons, such as for sea turtle populations, where females and sometimes males are not available for breeding every year and have offset breeding times within a season (Limpus, 1993; Davenport, 1997; Laloë et al., 2014; Schofield et al., 2017; Yaney-Keller, Martin & Reina, 2021; González-Cortés et al., 2021; Hays, Shimada & Schofield, 2022). Therefore, analyses for a single nesting season may not provide enough information to inform field sampling that will take place over several years to account for interannual variability. Moreover, we assumed that levels of multiple paternity and probabilities for breeding with certain numbers of mates remained constant, when they may vary as available adult density changes throughout a breeding season or between individuals (Wright et al., 2013; Lee et al., 2018; Yaney-Keller, Martin & Reina, 2021; González-Cortés et al., 2021). Both limitations could be addressed with probability distributions of breeding with multiple mates that include within or between season noise, or both, or are adjusted mechanistically based on density, overall population size, or the operational sex ratio, for example. However, all of these would require more population-specific data over longer timespans. As an age-based model, we were also unable to account for size-based reproductive success, or male preference for differently sized females, which has been shown in some sea turtle populations (Zbinden et al., 2007; Lasala et al., 2013; Lasala, Hughes & Wyneken, 2020; Das et al., 2025; Bispo et al., 2025), but see (Ekanayake et al., 2013; Wright et al., 2013; Sari, Koseler & Kaska, 2017; Howe et al., 2018; Lasala, Hughes & Wyneken, 2020; González-Cortés et al., 2021; Hatase, Watanabe & Kobayashi, 2024; Amorim et al., 2025). Finally, while this study focused on sample sizes of offspring and clutches, it assumed perfect information about genetic relationships without considering the effects of sampling eggs or hatchlings, alive or dead; the practice of collecting multiple samples per individual to account for potentially degraded samples; or complex analyses to balance sampling offspring with the number of loci, alleles, and statistical methods to be used for the reconstruction of paternal genotypes via kinship analysis (Vankan & Faddy, 1999; Dewoody et al., 2000; Neff, Repka & Gross, 2003; Croshaw, Peters & Glenn, 2009; Isberg, 2022).

Future field research attempting to calculate breeding sex ratios through parentage analysis should focus on sampling offspring from as many females and clutches as possible. Studies attempting to reach the same goal by quantifying mating systems in terms of operational sex ratios, probabilities of breeding with different numbers of individuals, and paternal contribution modes in the wild should be a lower, secondary priority. It is often assumed that male sea turtles will mate with multiple females, given anecdotal evidence of multiple copulations and the higher proportion of females that are often available for breeding, and genetic evidence supporting this has been reported in some sea turtle populations (Crim et al., 2002; Stewart & Dutton, 2014; Natoli et al., 2017; Gaos et al., 2018; Türkozan et al., 2019; Horne et al., 2023; Vella & Vella, 2024; Amorim et al., 2025; Bispo et al., 2025). However, many studies conducting parentage analyses on sea turtle populations did not detect any polygyny (Theissinger et al., 2009; Cruz et al., 2010; Joseph & Shaw, 2011; Stewart & Dutton, 2011; Wright et al., 2012b; Wright et al., 2012a; Ekanayake et al., 2013; Lasala et al., 2013; Phillips et al., 2013; Phillips et al., 2017; González-Garza et al., 2015; Alfaro-Núñez, Jensen & Abreu-Grobois, 2015; Sari, Koseler & Kaska, 2017; Chassin-Noria et al., 2017; Howe et al., 2018; Lasala, Hughes & Wyneken, 2018; Lasala, Hughes & Wyneken, 2020; González-Cortés et al., 2021; Silver-Gorges et al., 2024; Hatase, Watanabe & Kobayashi, 2024; Hennessey, 2025; Ohana et al., 2025), but sample sizes are typically small relative to clutch and population size, and parentage analysis has not been attempted at all for many populations, so the overall prevalence of polygyny in sea turtles is still largely unknown.

Conclusions

The objective of this study was to quantify the confidence in identifying all fathers that contribute to (i.e., sire) a single clutch and across clutches in a nesting season through parentage analysis for populations with multiple paternity. To quantitatively inform field sampling of offspring from individual clutches (within-clutch analysis) and from multiple clutches across nesting seasons (within-season analysis), we used computer simulations with demographic parameters based on a sea turtle (superfamily Chelonioidea) population to conduct power analyses and estimate the confidence of identifying all fathers given different population parameters, mating systems, and sample designs. Within clutches, confidence increased with marginal paternal contributions of the least dominant father, and increased, more weakly, with the number of offspring sampled. Results at this scale were dependent on sample sizes and knowing the correct paternal contribution mode, especially as the number of potential fathers increased, such that more research into the mating systems of species of interest would greatly increase the applicability of similar simulation results. Across clutches within a season, however, increasing the sample size of offspring within clutches did not increase confidence nearly as much as increasing the proportion of clutches sampled; simulations without polygyny required almost 100% of clutches to be sampled, since fathers were only represented in a single clutch throughout the season. However, relaxing the required proportion of fathers to identify from 100% to 90% greatly reduced the proportion of clutches that needed to be sampled to achieve a high confidence of 80% or higher, even for simulations without polygyny and much larger operational population sizes. Both within clutches and within seasons, increasing the sample size to more than one third of the offspring within a clutch would only be necessary in populations that have or are expected to have extremely skewed paternal contribution modes and many potential fathers, where the least dominant father contributes very little to clutch paternity. Therefore, sampling fewer offspring from as many clutches as possible is most likely the best way to maximize confidence in identifying all contributing fathers at both the clutch and nesting season scale. Our within-season results did not change substantially across scenarios with different paternal contribution modes, sample sizes of offspring within clutches, or mating systems, and are therefore applicable when estimating the breeding sex ratio in a wide array of animal populations whose mating systems may be generally unknown but include multiple paternity.

Supplemental Information

Supplemental Information 1 Females and clutches and eggs laid over four seasons on Praia do Leao, Fernando de Noronha, Brazil from 2020 through 2023

We collected these data via nightly nesting beach patrols from end of December through June of each nesting season. We identified nesting females through existing metal flipper tags or we designated new females and tagged them with metal flipper tags. Several days after nests hatched (or 60 days after the lay date, if hatching was not observed), we excavated all nests that could be found, and we calculated the total number of eggs as the sum of eggshell fragments (only counting eggshell fragments that were at least half an egg), dead and live hatchlings, and unhatched eggs. Data collection for the 2020 season was interrupted due to the COVID-19 pandemic, so we did not record any data past April 2nd. We calculated the distribution of the number of clutches per female based on data from the 2021–2023 nesting seasons, excluding clutches missing identified females but including clutches missing egg counts (Table 1). We calculated the distribution of the number of eggs per clutch based on data from all nesting seasons, excluding nests missing egg counts but including nests missing identified females and lay dates (Table 1). Distributions visually did not vary much between nesting seasons and were approximately normally distributed. All field work and data collection, including animal capture and tagging, was performed following protocols approved by the Institutional Animal Care and Use Committee at Florida State University (permits 1803, and PROTO202000076) and the Brazilian Ministry of Environment (MMA), Chico Mendes Biodiversity Conservation Institute (ICMBio), and Biodiversity Authorization and Information System (SISBIO), approval #69389-12.

Supplemental Information 2 Summary of literature quantifying multiple paternity in wild green turtle clutches

Values in each cell are the average proportion of green turtle clutches that had eggs fertilized by the F th father in clutches analyzed genetically. No study showed any wild green turtle clutches that had eggs fertilized by more than five fathers.

Supplemental Information 3 Probabilities of number of fathers identified in a clutch given the number of fathers contributing

Some rows do not appear to add up to 1.000 due to rounding.

Supplemental Information 4 Confidence in identifying all fathers by mating system and paternal contribution mode when 96 offspring are sampled from each clutch

Color indicates the upper edge of the confidence band, with solid fill representing simulations with the Random paternal contribution mode and lines representing simulations with the Dominant 90 paternal contribution mode. Within each panel, the x-axis represents the proportion of clutches across the entire nesting season that were sampled, and the y-axis represents the operational sex ratio. Columns of panels show different distributions of polyandry, with results from simulations with decreasing probabilities (“Decreasing polyandry” in panels (A), (B), and (C)) and uniform probabilities (“Uniform polyandry, in panels (D), (E), (F)) of females breeding with one to five males. Rows of panels show different distributions of polygyny, with results from simulations with decreasing probabilities of mating with 1–5 females (“Decreasing polygyny”, panels (A), (D)), uniform probabilities of mating with 1–5 females (“Uniform polygyny”, panels (B), (E)), and zero probabilities of males breeding with more than one female (“No polygyny”, panels (C), (F)).

Supplemental Information 5 Proportion of simulations that successfully identify enough fathers by population size and threshold of fathers identified with uniform polyandry and no polygyny

Color indicates the proportion of simulations with offspring sample sizes of 32 where the minimum required fathers were correctly identified, with solid fill representing simulations with the Random paternal contribution mode and lines representing simulations with the Dominant 90 paternal contribution mode. Within each panel, the x-axis represents the proportion of clutches across the entire nesting season that were sampled, and the y-axis represents the operational sex ratio. Columns of panels show different minimum proportions of fathers to ID to be counted as a “success”: 90% of all fathers (panels (A), (B), (C), (D) and 100% of fathers (panels (E), (F), (G), (H)). Rows of panels show different starting population sizes of adults available for breeding: 100 (panels (A), (E)), 200 (panels (B), (F)), 500 (panels (C), (G)), and 1,000 (panels (D), (H)). The mating system modeled had uniform probabilities of mating with one to five mates (“uniform polyandry” and “uniform polygyny”).

Thanks to Jonny Armstrong, GC Hays, Kostas Papafitsoros, and one anonymous reviewer for comments on manuscript drafts that improved relevance, accuracy, and clarity. VQ and WW conducted work and received institutional support on the stolen traditional homelands of the Yakina tribe of the Siletz Indians and the Mary’s River, or Ampinefu, band of the Kalapuya, whose descendants are alive today as part of the Confederated Tribes of Grand Ronde and the Confederated Tribes of Siletz Indians.

Additional Information and Declarations

Competing Interests

Author Contributions

Animal Ethics

Field Study Permissions

Data Availability

The authors declare there are no competing interests.

Vic Quennessen conceived and designed the experiments, performed the experiments, analyzed the data, prepared figures and/or tables, authored or reviewed drafts of the article, and approved the final draft.

Mariana M.B.P. Fuentes conceived and designed the experiments, authored or reviewed drafts of the article, and approved the final draft.

Lisa Komoroske conceived and designed the experiments, authored or reviewed drafts of the article, and approved the final draft.

J. Wilson White conceived and designed the experiments, authored or reviewed drafts of the article, and approved the final draft.

The following information was supplied relating to ethical approvals (i.e., approving body and any reference numbers):

All data collection, including animal capture and tagging, was performed following protocols approved by the Institutional Animal Care and Use Committee at Florida State University (permits 1803, and PROTO202000076) and the Brazilian Ministry of Environment−MMA (SISBIO/ICMBio, 69389-12).

The following information was supplied relating to field study approvals (i.e., approving body and any reference numbers):

Fieldwork on Praia do Leao in Brazil was approved by the Ministry of the Environment (MMA), Chico Mendes Biodiversity Conservation Institute (ICMBio), and Biodiversity Authorization and Information System (SISBIO), approval #69389-12.

The following information was supplied regarding data availability:

The code is available at GitHub and Zenodo:

– https://github.com/vquennessen/MultiplePaternityPowerAnalyses

– vquennessen (2025). vquennessen/MultiplePaternityPowerAnalyses: Initial release (v1.0.0). Zenodo. https://doi.org/10.5281/zenodo.14750961

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
