# Peer review of "Power analyses to inform clutch sampling design to determine the breeding sex ratio in populations with multiple paternity"

_PeerJ, doi:10.7717/peerj.20165_

## Round 0.1 · original submission · Major Revisions

Dear authors,

Please revise and provide a point-to-point reply to the questions raised by the different reviewers. Just so you know, reviewer 3 provides an annotated manuscript.

Best regards,

Fernando

Reviewer 1 ·

Basic reporting

no comment

Experimental design

not applicable

Validity of the findings

see section "additional comments"

Additional comments

The study is a valuable contribution that can help researchers assess the power of their sampling designs when estimating the number of sires in a cohort. In particular, this simulation study focuses on promiscuous species with large clutch sizes, such as sea turtles. While informative, the reliability and applicability of the study could be improved.

For similar simulation studies, it is crucial to carefully consider the range of parameters used to draw conclusions. These parameters should align with what is known or expected about the biology of the focal species. In this case, the simulations were partly based on unrealistic assumptions. Specifically, the authors assumed female polyandry but male monogamy. Additionally, the operational sex ratio (OSR) ranged from 0.05 to 0.5. While there are some, albeit weak (see below), arguments supporting male monogamy, no justification was provided for the chosen range of OSR. Similarly, authors did not provide any evidence supporting a population size of 100 adults. Population size is likely the most important parameter for a test aimed at identifying all sires. A higher number of males would require a larger sample size. If there is no evidence supporting a population size of 100 adults, the authors should extend their simulations to cover a range of population sizes.

Fisher’s principle of equal parental allocation to male and female offspring generally leads, through fitness being negatively dependent on frequency, to a 50:50 sex ratio, although this might be more variable in species with temperature-dependent sex determination, such as in turtles. Despite this potential variation, the default assumption should still be a balanced sex ratio unless empirical data suggest otherwise. Therefore, the null hypothesis should assume balanced ratio, and simulations should be centered around 0.5 rather than 0.28, as was done here (range: 0.05–0.5).

The unrealistic nature of some of these parameters is clearly demonstrated in Figure 1, where only 6 out of 100 females ready to mate and breed successfully found males and laid eggs, while 94 failed to do so. In a system where males do not invest in paternal care, such extreme female competition for mates is unrealistic. In reality, all females ready to breed are likely to find a mate, given that males invest only in sperm production. Importantly, we can predict this broad pattern from first principles of how sexual selection and parental care interact in the evolution of mating systems, although gathering more data on the natural history of sea turtles would be needed for more specific and quantitative predictions.

The authors cite four studies as support for male monogamy in sea turtles (Lee, Hays & Avise, 2004; Wright et al., 2012b; Ekanayake et al., 2013; Chassin-Noria et al., 2017). However, the first two studies neither test nor mention male monogamy. In fact, Lee et al. (2004) explicitly state: “Males will most certainly attempt to mate many times and fertilize the clutches of several different females.” The remaining two studies were not designed to test for male polygamy; they simply did not find direct evidence for it. However, this is a weak argument in favor of male monogamy. Both studies determined paternity for only a small sample of hatchlings from a limited number of clutches (16 and 24, respectively) collected over a year or two. Without knowledge of the total population size (which was not provided in either study), it is possible that the sampled hatchlings represent only a small fraction of the total population, resulting in low statistical power to detect male polygamy.

In summary, I think the study should focus on the most biologically plausible mating system: promiscuity in both males and females, along with a balanced sex ratio. The evidence against these theoretically expected patterns is, at best, weak. In the end, this adjustment may not significantly alter the results, as the authors also conducted simulations incorporating male polygamy, which yielded similar outcomes (lines 280–294 and supplemental figures). However, a focus on realistic parameters would enhance the rigor and reliability of the study.

Minor points:

Supplemental Figures lack captions.

Lines 170–172: There is no figure supporting this claim. Figure 1 presents different information.

Figures 3 and S1: These, along with the accompanying text (lines 238–252), seem redundant given the more informative Figure 2. Such redundancy and complexity may obscure rather than clarify conclusions.

Line 306: more than what?

Table 1: Published data for these parameters would be preferable. The source of these unpublished data is unclear—are they from the authors' own fieldwork?

·

Basic reporting

Here a power-analysis of performed to assess the sample sizes required to accurately identify the number of fathers for turtle clutches. This is a solid manuscript. I have some suggestions to improve the published version. It is fairly easy to run these sort of simulations to estimate confidence limits. But at times I wasn’t sure how the outcomes would help those trying to design a sampling programme … at time the authors seem to say that you need to know the answers to design the best sampling protocol. So I think a few minor corrections are needed so this manuscript offers more guidance and help to those aiming to work in this area.

1. Abstract.
“At the individual clutch level, we found that confidence was strongly dependent on the paternal contribution mode, and when the distribution of fertilized eggs among fathers was skewed, it also depended on the total number of fathers and the sample size.”

Some of these things will not be known when you design sampling. So what is the advice for people who do not know these things ?

2. Line 72.
“… copulation in green turtles is difficult to observe …”
First why limit the writing to green turtles and not sea turtles in general ?
Second, it is really difficult to observe ? In some places it is easy to spot mating pairs.
Eg. see Fig. 3 in:
Godley et al. Reproductive seasonality and sexual dimorphism in green turtles. MEPS Vol. 226: 125–133, 2002
Or Figs 2 and 3 in
Staines et al. Operational sex ratio estimated from drone surveys for a species threatened by climate warming. Marine Biology (2022) 169:152
https://doi.org/10.1007/s00227-022-04141-9

Do you mean it is difficult to identify, by direct observation, all the individual males a female mates with across the mating season ?


3. Line 70. “This is especially important given that green turtles are listed as globally endangered with a declining population trend (IUCN, 2024).”
This is misleading. They are increasing in abundance in many parts of the world and will shortly be listed as “least concern”.
See: Status, trends and conservation of global sea turtle populations. Nature Reviews Biodiversity, 1, 119–133. https://doi.org/10.1038/s44358-024-00011-y (2025)

I think a better justification is to say that even though the conservation status of green turtles is improving (e.g. above) concerns remain over feminisation of populations with one scenario being that adult males will consequently becomes limiting. A shortage of adult males is likely to be evidenced by a decrease in the incidence of multiple paternity, making accurate estimates of mp important. See:

Hays GC, Laloë J-O, Lee PLM, Schofield G (2023). Evidence of adult male scarcity associated with female-skewed offspring sex ratios in sea turtles. Current Biology 33, R1–R15. https://doi.org/10.1016/j.cub.2022.11.035


4. Lines 89-93. I would give your answers to these questions in the Abstract.

5. Line 118. “We capped the number of potential fathers at five …”
Per clutch ?

6. Line 122. “We similarly did not include polygyny, …”
Discuss the implications of this assumption. See also point 11.

7. Line 232 and line 235. “With a sample size of 32 …” “… sample size of 96…”
Perhaps specify “32 hatchlings per clutch …” etc.

8. Line 272. “Given the lack of polygyny …”
I would say “Given our assumption for a lack of …”

9. Lines 323. “Broadly speaking, to identify all the fathers in a single clutch, it would only make sense to sample more hatchlings in a clutch for animal populations with highly skewed paternal contribution modes where …”
This circles back to previous comments … are you saying you can only design a sampling programme if you already know the expected result ?

10. Line 296-345. This section of the Discussion is largely just repetition of Results. Rather, point-by-point throughout your Discussion, I would try and place your results in the context of the broader literature.

11. Line 394. “For green turtle populations specifically, polygyny was not detected …”
This statement is misleading. I think you need to write with an appreciation of the fact that in a big nesting population (let us say 10,000 females and 10,000 males), the chances that you detect polygyny if you sample 20 or 30 clutches is likely very small even if it occurs widely. It would be easy to show by modelling that the chances of detecting polygyny through parentage analysis will increase in small populations – and indeed this is where it has been shown in sea turtles.

In summary, a nice, solid manuscript that with some minor corrections will have more utility to those embarking in this area of research. Graeme Hays

Experimental design

See 1

Validity of the findings

See 1

Additional comments

See 1

·

Basic reporting

The paper is in general clear and well-understandable. Please see attached pdf for more details.

Experimental design

In general the design is meaningful. I have some remarks regarding the definitions of operational and breeding sex ratios and also regarding polygyny. Please see attached pdf for more details.

Validity of the findings

The results of the simulations are reasonable and as expected. Please see attached pdf for more details.

Additional comments

Please see attached pdf for more details.

---

## Round 0.2 · accepted · Accept

Congratulations. The reviewers are happy with your answers to the questions raised. I have no objections and can recommend the publication of the manuscript.

·

Basic reporting

The authors have made a good effort to revise the manuscript in line with the comments. Thank you to the authors for attending to the comment so thoroughly. I think this manuscript can now be accepted for publication in PeerJ. It will make a nice contribution. Graeme Hays

Experimental design

The authors have made a good effort to revise the manuscript in line with the comments. Thank you to the authors for attending to the comment so thoroughly. I think this manuscript can now be accepted for publication in PeerJ. It will make a nice contribution. Graeme Hays

Validity of the findings

The authors have made a good effort to revise the manuscript in line with the comments. Thank you to the authors for attending to the comment so thoroughly. I think this manuscript can now be accepted for publication in PeerJ. It will make a nice contribution. Graeme Hays

Additional comments

The authors have made a good effort to revise the manuscript in line with the comments. Thank you to the authors for attending to the comment so thoroughly. I think this manuscript can now be accepted for publication in PeerJ. It will make a nice contribution. Graeme Hays